# Improved Landscape Expansion Index and Its Application to Urban Growth in Urumqi

Yuhang Tian [1,2,3,4], Yanmin Shuai [2,3,4,*], Xianwei Ma [2], Congying Shao [2], Tao Liu [1,3,4] and Latipa Tuerhanjiang [1,4,5]

1 Xinjiang Institute of Ecology and Geography, Chinese Academy of Sciences, Urumqi 830011, China
2 College of Surveying and Mapping and Geographic Science, Liaoning Technical University, Fuxin 123000, China
3 College of Resources and Environmental, University of Chinese Academy of Sciences, Beijing 100049, China
4 CAS Research Center for Ecology and Environment of Central Asia, Urumqi 830011, China
5 College of Geography and Remote Sensing Sciences, Xinjiang University, Urumqi 830017, China
* Correspondence: shuaiym@ms.xjb.ac.cn

**Abstract:** Automatic determination of quantitative parameters describing the pattern of urban expansion is extremely important for urban planning, urban management and civic resource configuration. Though the widely adopted LEI (landscape expansion index) has exhibited the potential to capture the evolution of urban landscape patterns using multi-temporal remote sensing data, erroneous determination still exists, especially for patches with special shapes due to the limited consideration of spatial relationships among patches. In this paper, we improve the identification of urban landscape expansion patterns with an enhancement of the topological relationship. We propose MCI (Mean patch Compactness Index) and AWCI (Area-Weighted Compactness Index) in terms of the moment of inertia shape index. The effectiveness of the improved approach in identifying types of expansion patches is theoretically demonstrated with a series of designed experiments. Further, we apply the proposed approaches to the analysis of urban expansion features and dynamics of urban compactness over Urumqi at various 5-year stages using available SUBAD-China data from 1990–2015. The results achieved by the theoretical experiments and case application show our approach effectively suppressed the effects induced by shapes of patches and buffer or envelope box parameters for the accurate identification of patch type. Moreover, the modified MCI and AWCI exhibited an improved potential in capturing the landscape scale compactness of urban dynamics. The investigated 25-year urban expansion of Urumqi is dominated by edge-expansion patches and supplemented by outlying growth, with opposite trends of increasing and decreasing, with a gradual decrease in landscape fragmentation. Our examination using the proposed MCI and AWCI indicates Urumqi was growing more compact in latter 15-year period compared with the first 10 years studied, with the primary urban patches tending to be compacted earlier than the entire urban setting. The historical transformation trajectories based on remote sensing data show a significant construction land gain—from 1.06% in 1990 to 6.96% in 2015—due to 289.16 km$^2$ of recently established construction, accompanied by fast expansion northward, less dynamic expansion southward, and earlier extension in the westward direction than eastward. This work provides a possible means to improve the identification of patch expansion type and further understand the compactness of urban dynamics.

**Keywords:** LEI; patch compactness; topological relationships; remote sensing; Urumqi



## 1. Introduction

Automatic determination of urban expansion patterns is essential for urban planning, urban management and civic resource configuration. Cities and towns represent the epicenter in terms of politics, the economy, culture and population and maintain routine civic affairs of regions and countries. Accurate determination of the spatial distribution

and expansion features of different cities is an important prerequisite to understand the situation of the investigated cities and serves to support government decision making to optimize urban resources and improve urban ecology [1,2]. With economic development and residents' living requirements, global urbanization has increased with an enhanced velocity in recent decades. In 2021, about 56% of the global population lived in cities and towns, a rate expected to reach 68% in 2050 [3,4]. Compared with developed countries, with a rate higher than 80%, the percentage of urban dwelling in developing countries is as low as 20% [5]. China, a developing country, rapidly increased its percentage from 10.64% in the 1970s to 65% in 2021 [6]. Large populations continually aggregated in cities and towns, prompting the requirement of work positions, education, transportation and living facilities, hence inducing the fast expansion of urban areas. Issues thus frequently emerge from land cover changes, disturbance of the ecological environment and difficulty in the deployment of urban resources, with the process of urbanization [7–9]. Therefore, extracting urban expansion information and describing it with quantitative parameters is an urgent priority.

Numerous studies have focused on the evolution of urban spatial pattern dynamics to improve our understanding. The early investigation of urbanization mostly focuses on urban morphology. It examines the social or physical form of cities as well as human urban perceptions from sociology, economics, geography or psychology for improving the recognition of urban development and functioning mechanisms [10,11]. Subsequently, models constructed from metrology and system dynamics are frequently adopted to perform "top–bottom" empirical or mathematical statistical analyses of the relationship between urban morphology at different stages and various driving factors. The rapid development of satellite remote sensing has made it possible to analyze the structure, function and evolution rules of urban landscapes at the pixel scale. Cities and towns are considered as the scene-synthesis, consisting of different surface endmembers, used to explore urban dynamics qualitatively or semi-quantitatively. Next, ecological landscape parameters are introduced to describe the structure of urban landscapes [10]. For instance, the landscape index and landscape fragmentation are often used to reflect the components and configuration patterns of ecosystems, urban patches or functional units within the investigated region [12–14]. Further challenges are still met in accurately extracting spatial and temporal urban expansion information.

Models used to monitor the dynamics of urban landscape patterns have been updated with the development of remote sensing technology and increased requirements. The first urban growth model developed by the University of Connecticut can quantify the urban expansion and dynamics using land cover thematic maps [15]. Next, remote sensing–based urban process models have been widely applied to capture the landscape pattern changes of urban patches [16,17]. Three types of urban expansion patterns (*infilling*, *edge-expansion* and *outlying*) gradually emerged with the increasing accumulation of research on urban landscape change and dynamic evolution [18,19]. *Infilling* is defined as " the filling of non-urban area embraced by urban region or undeveloped land in urban built-up area" [20,21]. Liu et al. [18], based on the relationship among patches in remote sensing thematic maps, suggests that *infilling* is the pattern in which gaps (or holes) between or within old patches are filled with newly grown patches. *Edge-expansion* is considered as a pattern of the gradual expansion outwards from the urban fringes. *Outlying* is the expansion pattern in which the new patch has no adjacent relationship with existing patches. In the urban planning community, *infilling* can mostly increase the rate and compactness of urban land use, especially for the utilization of in-city undeveloped land [22,23]. The bid rent theory indicates that the cost decreases with increasing distance away from the center of the urban district [24]. Although both *infilling* and *edge-expansion* patterns increase the compactness of urban morphology, the cost of *edge-expansion* is lower than *infilling*. Therefore, accurate extraction of expansion patterns is a prerequisite for evaluating urban dynamic features.

The accumulation of multiple urban landscape data has made it possible to develop related models and quantitatively capture urban landscape pattern changes. Two kinds of

indices, named as reference-based and non-reference models, are designed to describe the dynamic changes in urban landscape patterns. Reference-based methods frequently adopt empirical thresholds to categorize urban expansion patterns, such as using the ratio of common boundary lengths between new and old patches to the perimeter of new patches [25]. The Landscape Expansion Index (*LEI*) is defined in terms of two empirical parameters, the patch envelope box ($LEI_{box}$) and patch buffer ($LEI_{buffer}$), to aid quantifying the situation of urban expansion patterns [18]. Evaluation of the urban compactness over Dongguan has captured the major features at the landscape scale using the modified *LEI*, Mean Expansion Index (*MEI*) and Area-Weighted Mean Expansion Index (*AWMEI*) [18,26]. The Proximity Expansion Index (*PEI*) is presented to enhance the ability in capturing the newly added outlying patches, depending on the distance between new and old patches and the shared ratio of boundaries, followed by a special emphasis in determining the parameters of buffer distance [27]. Jiao et al. [27] pointed out that it is more reasonable to set the buffer distance as the average scale of geographic entities that separate urban neighborhoods, such as the width of urban roads. The shape-weighted landscape evolution index (*SWLEI*) uses image pixel neighborhoods to quantify bidirectional changes in urban landscapes and to improve the ability to identify the spatial expansion patterns of specially shaped patches [28]. Xia et al. [28] pointed out that *LEI* is susceptible to the effects of patch shape and buffer distance. The non-reference method generally adopts topology relationships between old and new patches to identify urban expansion patterns. Wu et al. [29] defined a new landscape expansion index ($LEI_w$) according to the topographic relationship between new and old patches to recognize the adjacent expansion or exterior expansion. A further effort is performed to distinguish the relationship of "inclusion", "intersection", and "disjoint" between new and old patches [23]. Ghani and Abidin [22] modified the definition of $LEI_{buffer}$ depending on the detection of "inclusion" and "intersection" between the buffers of new and old patches to reduce the misclassification of *infilling* and *edge-expansion* patterns. The Landscape-Adjacency Index (*LAI*) was proposed to measure the regional ecological pressure of urban sprawl on natural landscapes based on the spatial adjacency between urban landscape patches and natural landscape patches [30].

Various LEIs exhibit different potential in identifying patterns of urban expansion patches as that shown for the three LEIs widely used, as shown in Table 1. These indices have made efforts to help improve the accuracy of capturing the dynamic information of landscape patterns at different periods, while limitations still exist with the requirements of these applications, as follows: (1) Parameters of the reference-based method, such as the buffer distance or the expansion multiple of the envelope box, can affect the expanded patch area, original patch area and blank area within the reference object, which may increase the uncertainty of the *LEI*. (2) $LEI_{box}$ and $LEI_{buffer}$ methods have detectable uncertainty, especially for patches with multilateral shape. (3) Non-reference methods such as $LEI_w$ can avoid the issues in the above two items due to its non-reference, but only two landscape expansion patterns can be identified. In addition, $LEI_{buffer}$ and $LEI_{box}$ enable the quantitative estimation of the compactness of urban expansion at the landscape scale, while $LEI_w$ does not.

To address the above issues, we aimed to upgrade the identification method of landscape expansion pattern depending on the existing topological relationship among patches. Further, a stable approach is modified from the classical one for the evaluation of urban compactness. The city of Urumqi, which has undergone a dramatic urban evolution, is chosen as an example to demonstrate the effectiveness of our methodology. Our study enriches the methodological system of landscape pattern index research, and the relevant conclusions are available to government policy makers to manage urban sprawl optimally.

**Table 1.** Comparison of three different landscape expansion indexes.

| | $\text{LEI}_{\text{box}}$ | $\text{LEI}_{\text{buffer}}$ | $\text{LEI}_{\text{w}}$ |
|---|---|---|---|
| Formula | $LEI_{box} = \begin{cases} 100 * \frac{A_O}{A_E - A_N} \text{ new patch is not rectangular} \\ 100 * \frac{A_{LO}}{A_{LE} - A_N} \text{ new patch is rectangular} \end{cases}$ | $LEI_{buffer} = 100 * \frac{A_c}{A_c + A_d}$ | $LEI_w = \frac{A_n - A_o}{A_n + A_o}$ |
| Parameter | $A_E$ is the envelope box area of the new patch, $A_N$ is the area of the new patch, $A_O$ is the area of the old patch inside the new patch envelope box, $A_{LE}$ is the expanded envelope box area of the new patch, $A_{LO}$ is the area of the old patch inside the new patch expanded envelope box. | $A_C$ is the overlap area between the new patch buffer and the old patch, $A_d$ is the difference between the buffer area and $A_C$. | $A_n$ is the area of new patches, and $A_o$ is the area of patches that are adjacent to the new patches. |
| Rationale | Envelope box based on new patch | Buffer based on new patch | Based on the adjacency of the new patch to the old patch |
| Value range | [0, 100] | [0, 100] | (−1, 1] |
| Influence factors | New patch shape, envelope box expanded multiples | New patch shape, buffer distance | Nothing |
| Expansion pattern | (50, 100], *infilling* [2, 50], *edge-expansion* [0, 2), *outlying* | (50, 100], *infilling* (0, 50], *edge-expansion* 0, *outlying* | (−1, 1), *adjacency expansion* 1, *external expansion* |

## 2. Methodology

### 2.1. Effective Identification of Landscape Expansion Patterns

Urban landscape expansion patterns can be classified as *infilling*, *edge-expansion* and *outlying*. In Figure 1a–c, *infilling* refers to the type where undeveloped patches (holes) in built-up urban areas are filled with newly grown urban patches; in Figure 1d, *edge-expansion* refers to the kind that extends outward from the edge of the urban built-up area; in Figure 1e *outlying* refers to the type of new urban patches that are separated from the built-up area.

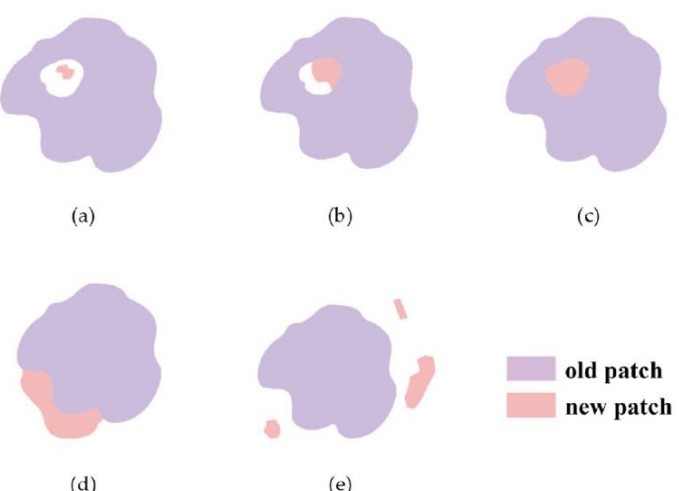

**Figure 1.** Schematic diagram of the three landscape expansion patterns. (**a**–**c**) represents *infilling* growth, (**d**) represents *edge-expansion* growth, and (**e**) represents *outlying* growth.

This paper improves the identification of urban landscape expansion patterns with an enhancement of the topological relationship among patches. In Figure 2, the new urban patches and the old urban patches are available by the difference analysis of built-up area vector boundaries in the period before and after; the holes inside the built-up area vector can be accessed through the topology examination function. These patches are input to the topological relationship analysis module. The key to identification of landscape expansion patterns is to analyze the topological relationships among the old patches, the holes and the new patches. Specifically, if the holes contain or overlap the new patch, the new patch

is identified as *infilling*; if the new patch is disjointed from the hole and touches to the original patch, the new patch is identified as an *edge-expansion*; and if the new patch is disjointed from the hole and old patch, the new patch is identified as *outlying*. In this paper, the analysis of spatial topological relationship is implemented based on the Python 3.8 GeoPandas project, and the hole identification is implemented based on the ArcGIS 10.6 topology examination function.

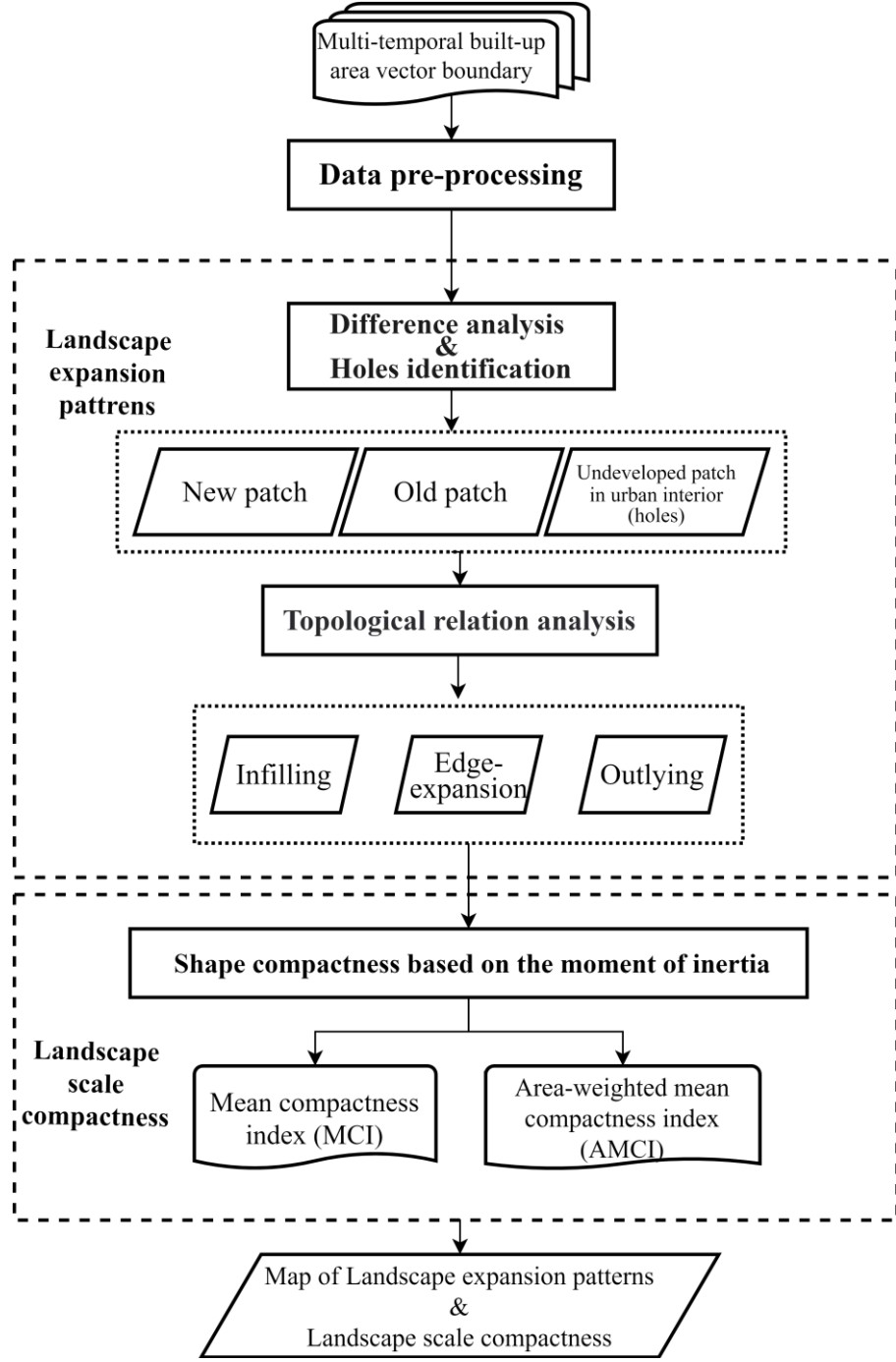

**Figure 2.** Process framework for landscape expansion pattern identification and compactness calculation.

### 2.2. Comparison Verification

To verify the effectiveness of the improved landscape expansion pattern identification method, several specific shape patches were selected to compare the improved method with the reference-based landscape expansion pattern identification method.

Figure 3a,b belong to the envelope box method, and the calculation results are in Table 2. According to the landscape expansion pattern identification rules of the envelope box method, the new patch in case (a) with an LEI equal to 100 is identified as *infilling*, which is actually an *edge-expansion*, expanding outward along the fringe of the built-up area. The new patch can be correctly identified by topological touch with the old patch. The new patch in case (b) with an LEI equal to 27.45 is identified as *edge-expansion*, which is an *outlying*, and can be correctly identified by new patch disjoint from the old patch.

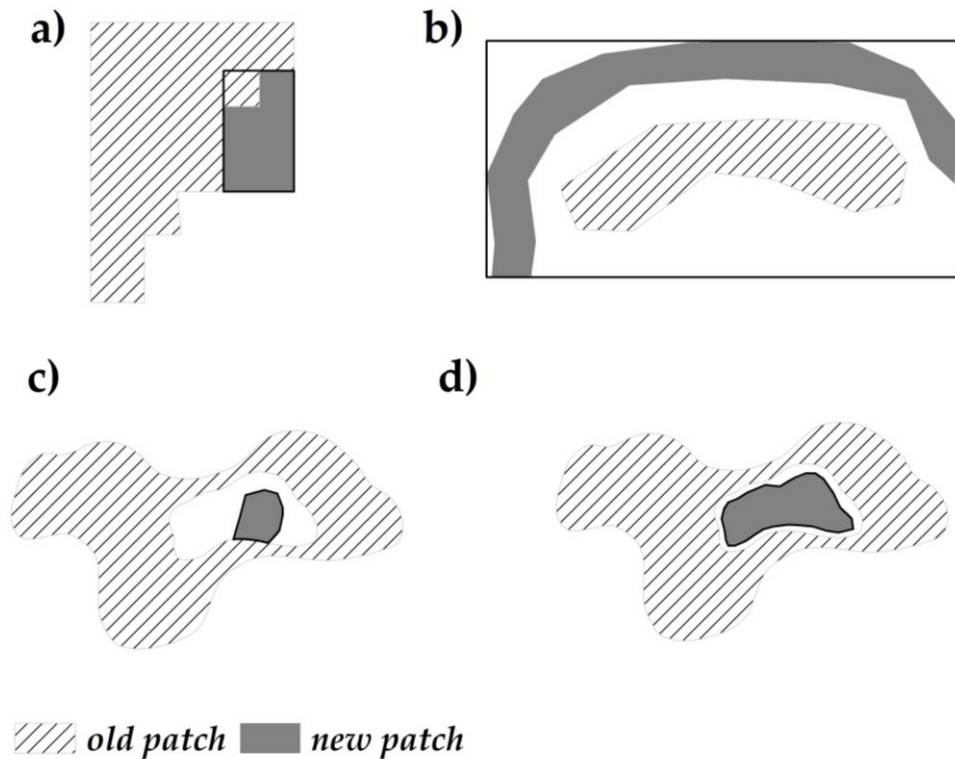

**Figure 3.** Cases of misclassification of landscape expansion index based on the envelope and the buffer method. (**a**,**b**) is the envelop box method. (**c**,**d**) is the buffer method.

**Table 2.** Calculation results of the envelope box method. Area unit: km$^2$.

| Case | Envelope Area ($A_E$) | New Patch Area ($A_N$) | Old Patch in Envelope ($A_O$) | LEI |
|------|------------------------|-------------------------|-------------------------------|-----|
| (a) | 151.60 | 128.91 | 22.69 | 100 |
| (b) | 1989.34 | 459.30 | 420.05 | 27.45 |

Figure 3c,d belongs to the buffer method, and the buffer distance is set to 1 m; Table 3 describes calculation results. According to the landscape expansion pattern identification rules of the buffer method, the new patch in case (c) with an LEI equal to 20.93 is identified as *edge-expansion*, which is actually the *infilling* of the interior of the built-up area of the city; similarly, case (d), with an LEI equal to 0, is identified as *outlying*, which is actually also *infilling*. Both can be correctly identified by the new patch overlap with the hole.

**Table 3.** Calculation results of the buffer method. Area unit: m$^2$.

| Case | Overlap Area between Buffer and Old Patch ($A_C$) | Overlap Area between Buffer and Blank Area ($A_d$) | LEI |
|---|---|---|---|
| (c) | 47.33 | 178.80 | 20.93 |
| (d) | 0 | 455.49 | 0 |

In addition to the above cases, the buffer-based LEI is influenced by the buffer distance; the final LEI is calculated based on the buffer distance of 1 m to obtain a robust outcome. The magnification of the envelope also influences the envelop method. However, the buffer distance for different cities needs to be optimized by experimentation, and there is limited generalizability of setting it to 1 m [27]. Figure 3d represents filling interior urban gaps; however, different buffer distances will result in different expansion patterns. A too-small buffer distance will be identified as *outlying*, while a buffer distance above a specific value will be identified as *infilling*. In summary, the improved landscape extension pattern identification method depends on the existing spatial topological relationship among patches, which is not influenced by the shape of the patch and the reference object parameters, thus solving the misclassification problem to a certain extent.

### 2.3. Calculation of the Landscape Scale Compactness

To better understand the trend of urban morphological changes, we introduced the moments of inertia shape index and calculated the landscape scale compactness of the urban area with the developed variants *MCI* and *AWCI* [31]. The landscape scale urban compactness reflected the changes in urban morphology due to the changes in the three landscape expansion patterns, which is an effective complement and validation of the results of landscape expansion pattern identification. The isoperimetric quotient ($C_{IPQ}$) is a commonly used compactness algorithm, which is the ratio of the area of a patch to the area of a circle patch with the same perimeter (the tightest patch) and is calculated by the formula in Equation (1).

$$C_{IPQ} = \frac{4\pi A}{P^2} \tag{1}$$

where $A$ is the area of the patch, $P$ is patch's perimeter, and $C_{IPQ}$ is the patch compactness index, which ranges from (0, 1) and is equal to 1 only if the patch is a circle. It is easily calculated and insensitive to size change, but the compactness algorithm is not robust enough to handle irregularly contoured patches and patches with holes [31,32].

The moment of inertia shape index has the following advantages over the conventional isoperimetric quotient method: (1) It can easily deal with holed and multi-part elements and (2) it has robustness in dealing with irregular boundary patches [31]. These advantages made it possible to quantify the difference in compactness that the holes bring to the overall patch when they are filled and also allow quantification of robust compactness for expansion patches with different shapes. The moment of inertia shape index is calculated as the ratio of the moment of inertia of a circle with same area about its center, to the moment of inertia of the shape about its centroid; see Equation (2). However, the compactness of the above index is only represented at the scale of individual patches, so we defined *MCI* and *AWCI* based on the moment of inertia shape index to reflect the compactness changes caused by urban expansion at the landscape scale; the formulae are given in Equations (3) and (4).

$$C_{MI} = \frac{A^2}{2\pi I_g} \tag{2}$$

where, $A$ is the area of the patch, $I_g$ is the moment of inertia of the patch about its centroid, and $C_{MI}$ is the shape index based on the moment of inertia, ranging from (0, 1), with higher values meaning that the patch is more compact.

$$MCI = \sum_{i=1}^{N} \frac{C_{MI_i}}{N} \tag{3}$$

where $MCI$ is the mean compactness index of each urban patch in a certain period, $N$ is the number of patches, and $C_{MIi}$ is the moment of inertia shape index of the patch $i$. The higher the $MCI$, the more compact the urban landscape tends to be in that period.

$$AWCI = \sum_{i=1}^{N} C_{MI_i} \times \frac{a_i}{A} \tag{4}$$

where $AWCI$ is the area-weighted compactness of each urban patch in a certain period, $a_i$ is the area of patch $i$, $A$ is the total area of urban patches during that period, and $C_{MIi}$ is the moment of inertia shape index of the patch $i$. $AWCI$ further explains the changes in compactness brought about by main urban patches.

In addition, the *Largest Patch Index* (*LPI*) indicates the percentage of area occupied by the largest patches of urban built-up land. Increasing *LPI* means the expansion of urban cores and can be used to supplement the explanation of changes in urban compactness caused by main patches.

$$LPI = \frac{MAX(a_i)}{A} \times 100 \tag{5}$$

where $MAX(a_i)$ is the area of the largest patch in a period, and $A$ is the total area of the patch in that period.

## 3. Application over Urumqi

In this paper, we applied the improved method to identify the landscape expansion patterns of different patches and calculated the compactness of urban patches in each period, to objectively evaluate the spatial and temporal expansion characteristics of the city. For our work, we chose the period between 1990 and 2015 to understand the urban expansion process of Urumqi city.

### 3.1. Study Area and Data

The city of Urumqi, located in northwest China, is the largest city in Central Asia. It currently has seven districts and one county under its administration, with mountains in its southwest and northeast regions as well as the alluvial plains of the Urumqi river and Toutunhe river in the north. The total area of Urumqi is approximately 14,577 km$^2$, whose impervious surface increased from 51.61 km$^2$ to 340.25 km$^2$ between 1990 and 2015, an increase of 559.27%. Urumqi is a typical city experiencing dramatic landscape pattern evolution; the specific location of the city is shown in Figure 4.

The data used in this paper mainly include the following: (1) The standardized urban built-up area dataset (SUBAD) of Urumqi for the five 5-year time periods involving the years 1990, 1995, 2000, 2005, 2010 and 2015 was selected as the basic data. The dataset was extracted from multi-source remote sensing data (time-series of Landsat and Sentinel images), and vectorized data were provided. The address of the dataset is available at http://www.doi.org/10.11922/sciencedb.j00076.00004, accessed on 1 August 2022 [33]. The years between 1990–2015 were chosen because the dataset was not developed for the 2020 built-up area data. After the difference analysis of the built-up area patches in the two periods before and after, the new patches in each time period were captured respectively. The spatial distribution of the new patches of urban built-up land in the study area between 1990 and 2015 is shown in Figure 4. (2) The urban built-up areas in 1990, 1995, 2000, 2005, 2010 and 2015, obtained from data from the Ministry of Housing and Urban-Rural

Development of the People's Republic of China; the address of the data is available at http://www.mohurd.gov.cn, accessed on 1 August 2022.

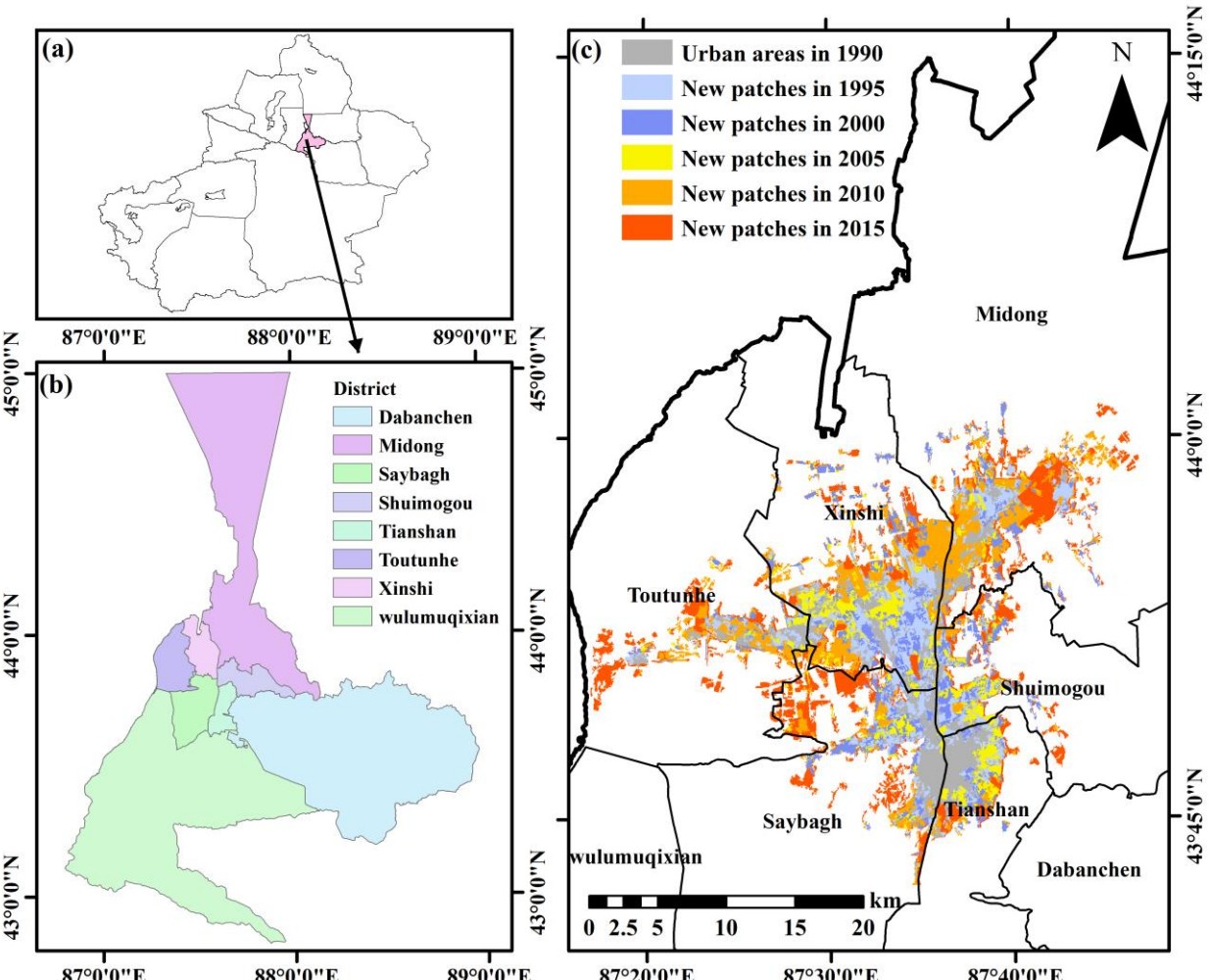

**Figure 4.** Overview map of the study area. (**a**,**b**) The spatial location of the study area. (**c**) The spatial distribution of expansion patches in Urumqi from 1990–2015.

### 3.2. Landscape Expansion Pattern

We extracted the landscape expansion patterns of urban expansion patches in Urumqi for five periods from 1990, 1995, 2000, 2005, 2010 and 2015. Figure 4c shows the spatial distribution of new patches in the built-up area of Urumqi city from 1990 to 2015. The core district of Urumqi in 1990 was mainly located in the southern Tianshan district and the northeastern part of the Saybagh district, while smaller patches were scattered in the Xinshi district, Tutunhe district, Shuimogou district and Midong district. The urban built-up area extended to the north and expanded in the east–west direction every five years between 1990 and 2015; the urban area of Urumqi in general formed a "T" shape with a long north–south direction and a narrow east–west direction in 2015.

Figure 5a shows the number percentage of the three landscape expansion patterns in the five periods. An overview of the five periods shows that the number percentage of *edge-expansion* is greater than *outlying*, and *outlying* is greater than *infilling*, among which the number of *edge-expansion* patches percentage was 87.77%, 80.01%, 81.34%, 84.64% and 88.08% in the five periods, showing a decreasing trend followed by an increasing trend. The number of *outlying* patches percentage was 9.65%, 14.08%, 11.71%, 11.51%, and 10.51% in the five periods, with a sharp increase from 1995 to 2000 and an overall trend of first increase and then decrease. The percentage of the number of *infilling* patches was 2.58%,

5.83%, 6.95%, 3.85%, and 1.41% in the five periods, showing an overall trend of first increase and then decrease. Figure 5b shows the area percentage of the three landscape expansion pattern patches in the five periods. *Edge-expansion* accounted for 84.69% of the total area of new patches in 1990–1995, *outlying* accounted for 13.99% of the total area, and *infilling* accounted for the least amount of area at 1.32%. The *edge-expansion* patches area percentage decreased significantly to 53.95% from 1995 to 2000; on the contrary, the *outlying* and *infilling* patches area percentage increased significantly to 35.34% and 10.71%, respectively. The *infilling* patches area percentage continued to increase to 17.28% from 2000 to 2005, which started to be higher than the *outlying* patches area percentage, while the percentage of *edge-expansion* patch area also increased to 75.44%; on the contrary, the percentage of *outlying* patch area decreased to 7.28%. The area percentage of the three landscape expansion patterns from 2005 to 2010 was 5.80% for *infilling*, 80.76% for *edge-expansion*, and 13.44% for *outlying*. The area percentage of the three landscape expansion patterns from 2010 to 2015 was 2.42% for *infilling*, 75.23% for *edge-expansion*, and 22.33% for *outlying*.

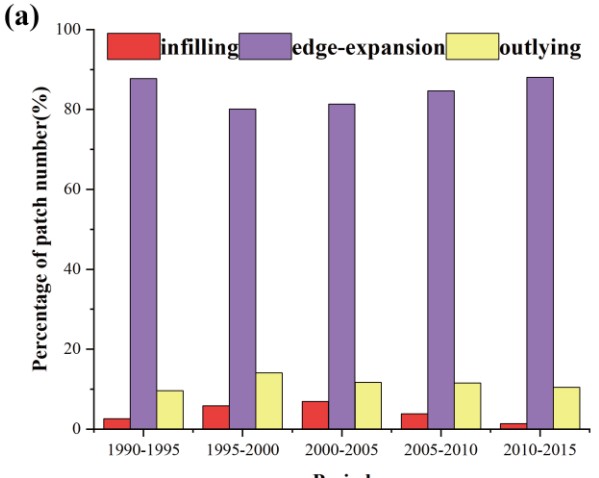 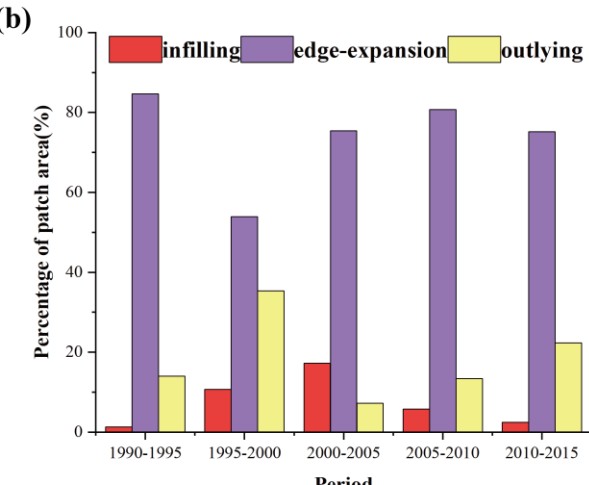

**Figure 5.** Percentage of area and number of patches with different landscape expansion patterns. (**a**) Percentage of the number of different patches. (**b**) Percentage of area of different patches.

Table 4 shows the overall number and area of the three landscape expansion patterns over the five 5-year periods extracted using the improved method. The number and area of *edge-expansion* patches were consistently higher than *outlying* and *infilling* patches in all time periods. The number of *outlying* patches was higher than that of *infilling* patches in all five time periods, but the area of *outlying* patches was smaller than that of *infilling* patches for the first time between 2000–2005.

**Table 4.** Increment of three landscape expansion patterns between 1990–2015.

| Years | Number of New Patches | | | | Area of New Patches (km²) | | | |
|---|---|---|---|---|---|---|---|---|
| | Infilling | Edge-Expansion | Outlying | Total Number | Infilling | Edge-Expansion | Outlying | Total Area |
| 1990–1995 | 34 | 1155 | 127 | 1316 | 0.68 | 43.71 | 7.22 | 51.61 |
| 1995–2000 | 92 | 1263 | 222 | 1577 | 5.63 | 28.36 | 18.58 | 52.57 |
| 2000–2005 | 118 | 1382 | 199 | 1699 | 6.41 | 27.99 | 2.70 | 37.1 |
| 2005–2010 | 86 | 1890 | 257 | 2233 | 4.24 | 59.05 | 9.83 | 73.12 |
| 2010–2015 | 21 | 1308 | 156 | 1485 | 2.07 | 64.28 | 19.08 | 85.44 |

Figure 6 shows the spatial distribution of different landscape expansion pattern patches in Urumqi during five periods from 1990 to 2015. Incorporating Table 4 and Figure 6, there were 1316 expansion patches in Urumqi between 1990 and 1995, and 34 patches belonged to *infilling*, with the total area of 0.68 km$^2$, mainly located in the Saybagh and Tianshan districts. There were 127 patches belonging to *outlying*, mainly located in the towns near Midong and Xinshi districts, totaling about 7.22 km$^2$. *Edge-expansion* accounts for the majority, 1155 patches in total, mostly located in the southern part of the Xinshi district, Tianshan district and Saybagh district, totaling about 43.71 km$^2$. There were 1577 expansion patches from 1995 to 2000, and 222 belonged to *outlying*, with a total area of about 18.58 km$^2$, mainly located in the Xinshi district, Midong district and the northern part of the Tianshan district. The *outlying* and *infilling* patches were much higher in both area and number than in the previous period. There were 92 *infilling* patches with a total area of about 5.63 km$^2$, mostly located in the Xinshi district, Saybagh district and the north of Tianshan district. The number of *edge-expansion* patches increased slightly and the area decreased significantly during this period, but *edge-expansion* was still the main expansion pattern during this period and was widely distributed, mainly located in districts other than Wulumuqi county and the Durban district. The number of new patches increased but the area decreased significantly between 2000 and 2005. Among these patches, there were 199 *outlying* patches with a total area of about 37.1 km$^2$, scattered in Saybagh district, Xinshi district and Midong district. There were 118 *infilling* patches with a total area of about 6.41 km$^2$, mainly located in the Saybagh district, Xinshi district and Tianshan district. There were 1382 *edge-expansion* patches with a total area of about 27.99 km$^2$, with the *edge-expansion* patches in this period mainly located in the Xinshi district, Shuimogou district and Tianshan district. Between 2005 and 2010, there were 2233 expansion patches with a total area of about 73.12 km$^2$, and the number of expansion patches reached the peak of the five periods. Among these patches, there were 1890 *edge-expansion* patches with a total area of 59.05 km$^2$, representing the majority, mainly in the Toutunhe district, Xinshi district and Midong district. There were 257 *outlying*, with a total area of 9.83 km$^2$, mainly in the outskirts of the urban areas of the Midong district and Toutunhe district. The *infilling* expansion consisted of 86 patches with a total area of 4.24 km$^2$, mainly in the outskirts of the Saybagh district, Shuimogou district and Tianshan district. Between 2010 and 2015, there were 1485 expansion patches with a total area of about 85.44 km$^2$, and the area of expansion patches reached the peak of the five periods. Among these patches, there were 1308 *edge-expansion* patches with a total area of 64.28 km$^2$, mainly in the Midong district. There were 156 *outlying* patches with a total area of 19.08 km$^2$, mainly in the Toutunhe and Midong districts. The *infilling* expansion consisted of 21 patches with a total area of 2.07 km$^2$, mainly in the south of the Xinshi district.

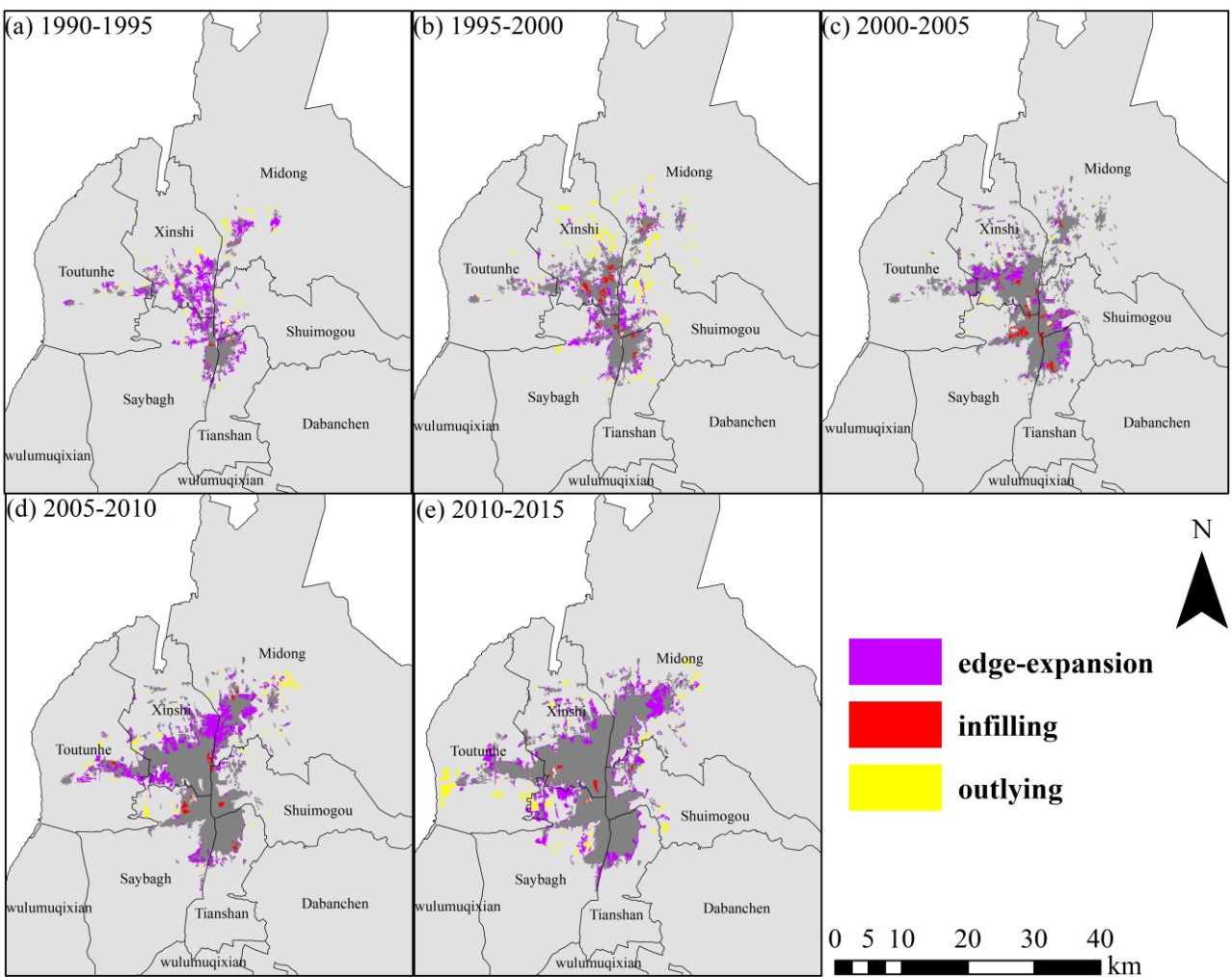

**Figure 6.** Spatial distribution of landscape expansion patterns in Urumqi from 1990 to 2015. (**a–e**) denote the spatial distribution of landscape expansion patterns for each of the five 5-year intervals from 1990–2015.

### 3.3. Landscape Scale Compactness

In Table 5, we calculated the *MCI* and *AWCI* of Urumqi for each period between 1990 and 2015 using Equations (3) and (4), respectively, to capture the compactness of urban expansion at the landscape scale. *MCI* showed a significant trend of decreasing and then increasing between 1990 and 2015, and it can be observed that 2000 was a turning point; the urban expansion after 2000 gradually tended to be compact and reached a peak of the five periods in 2015. The trend of *AWCI* is generally consistent with *MCI*, but there was a sharp increase between 1995 and 2000, in contrast to *MCI* showing a decreasing trend. The maximum connectivity patch compactness index (*MAX-CM*) is the moment of inertia shape index of the patch with the largest area, which is calculated as in Equation (2). The maximum connectivity patch area proportion *LPI* was further calculated for each period according to Equation (5). The maximum connectivity patch area for each period was also counted. The results showed that the change trends of *MAX-CM* and *AWCI* were highly consistent. In addition, the maximum connectivity patch area continued to increase, and *LPI* was consistent with this trend, except for 2015.

**Table 5.** Landscape scale compactness and related indexes in Urumqi between 1990 and 2015.

|  | **1990** | **1995** | **2000** | **2005** | **2010** | **2015** |
|---|---|---|---|---|---|---|
| *MCI* | 52.39 | 52.29 | 51.63 | 52.72 | 53.77 | 55.39 |
| *AWCI* | 50.85 | 39.19 | 49.69 | 54.47 | 52.46 | 52.08 |
| *MAX-CM* | 52.98 | 32.15 | 49.84 | 55.92 | 52.63 | 52.17 |
| *MAX-Area* (km$^2$) | 23.07 | 62.70 | 95.97 | 138.79 | 220.54 | 261.17 |
| *LPI* | 44.69 | 60.75 | 61.61 | 71.96 | 83.08 | 76.72 |

*3.4. Urban Historical Transformation Trajectories and Expansion Direction*

Previous studies have shown that describing historical transformation trajectories of land cover (including built-up land) is a fundamental step in the rational planning of landscape patterns [34,35]. To examine the changes that occurred during the investigated time interval, we used a post-classification comparison approach for historical analysis [36]. Two vectors of construction land data (1990–2015) derived from SUBAD were overlaid in a GIS environment, obtaining a unique vector that responds to the trajectory of construction land change. We built a complete transfer matrix to quantify the change in built-up land in the main urban zone (except for Wulumuqi county and the Durban district) of Urumqi, reporting in rows the value of changes in the 1990 category and in columns the number of changes in the 2015 category.

In Table 6, The built-up land increased from 51.61 km$^2$ in 1990 to 340.25 km$^2$ in 2015, representing a rise of the proportion of construction land to the main urban area from 1.06% to 6.96%. The overall trend observed shows an absolute increase of 559.27% (288.64 km$^2$) in the built-up area land cover, 5.9% if referring to the main urban zone area growth. Overall, there was a built-up land gain of 289.16 km$^2$, against a cover loss of 0.52 km$^2$. Built-up cover persistence involved 51.09 km$^2$ of construction land over the total built-up area of the main urban area of Urumqi.

**Table 6.** Contingency matrix showing the transition between different built-up and non-built-up land cover (values in km$^2$) from 1990 to 2015 in the main urban zone of Urumqi.

| **Built-Up/Non-Built-Up Land Cover** | **2015** | | |
|---|---|---|---|
|  | **Non-Built-Up** | **Built-Up** | **Total** |
| 1990 |  |  |  |
| Non-built-up | 4548.47 | 289.16 | 4837.63 |
| Built-up | 0.52 | 51.09 | 51.61 |
| Total | 4549.00 | 340.25 | 4889.25 |

To further understand the spatial and temporal patterns of urban development, we calculated the expansion area of the study area in eight major directions from 1990–2015 using zonal statistics. Table 7 shows the area statistics of the built-up area of Urumqi in the eight expansion directions of east, west, south, north, southeast, northwest, northeast and southwest for the five 5-year periods from 1990 to 2015. The main urban expansion directions during 1990–1995 were north, northwest and northeast, while the other new patches were primarily located in the south and southeast directions. The main expansion direction remained stable during 1995–2000, and the new area added in the northeast direction started to be larger than that in the northwest direction, while the expansion in the south and southeast was restrained, and the growth in the southwest direction increased. The development of the city slowed down in most directions between 2000 and 2005, but the northwest direction remained on a continuous expansion trend. Urban expansion accelerated significantly between 2005 and 2010, with north, northeast and west becoming the new growth poles for urban development, and the remaining expansion area concentrated in the northwest. The urban expansion continued during 2010–2015, with the west and northeast of the three growth poles remaining the main expansion direction. The northern expansion significantly decreased, but remained a vital expansion

direction. According to Figure 7, the southwest and southeast regions of Urumqi are at a higher elevation; urban expansion area is lower in all five periods. The northern alluvial plain is under the main direction of expansion, being the most significant metropolitan expansion area in the 1990–2010 period and showing an increasing trend, remaining an important expansion direction in 2015. The western region has significantly increased its expansion area since 2005, becoming another major growth pole. The area of expansion in the southern region was relatively stable and at low values in all five periods. With the incorporation of Miquan city into Urumqi in 2007 to establish the Midong district, the city has been characterized by a significant expansion to the northeast.

**Table 7.** Area of urban expansion in eight directions in Urumqi from 1990–2015. Area unit: km$^2$.

|           | 1990–1995 | 1995–2000 | 2000–2005 | 2005–2010 | 2010–2015 |
|-----------|-----------|-----------|-----------|-----------|-----------|
| West      | 4.62      | 3.46      | 2.56      | 14.68     | 25.10     |
| Southwest | 2.39      | 4.73      | 1.37      | 3.22      | 5.19      |
| South     | 5.51      | 4.28      | 3.23      | 4.37      | 7.80      |
| Southeast | 6.75      | 5.88      | 7.57      | 1.64      | 3.67      |
| East      | 2.09      | 3.74      | 3.19      | 0.55      | 2.84      |
| Northwest | 8.82      | 7.28      | 8.63      | 9.95      | 5.99      |
| Northeast | 7.78      | 10.19     | 3.77      | 16.89     | 23.44     |
| North     | 13.65     | 13.00     | 6.77      | 21.81     | 11.42     |
| Sum       | 51.61     | 52.56     | 37.09     | 73.11     | 85.45     |

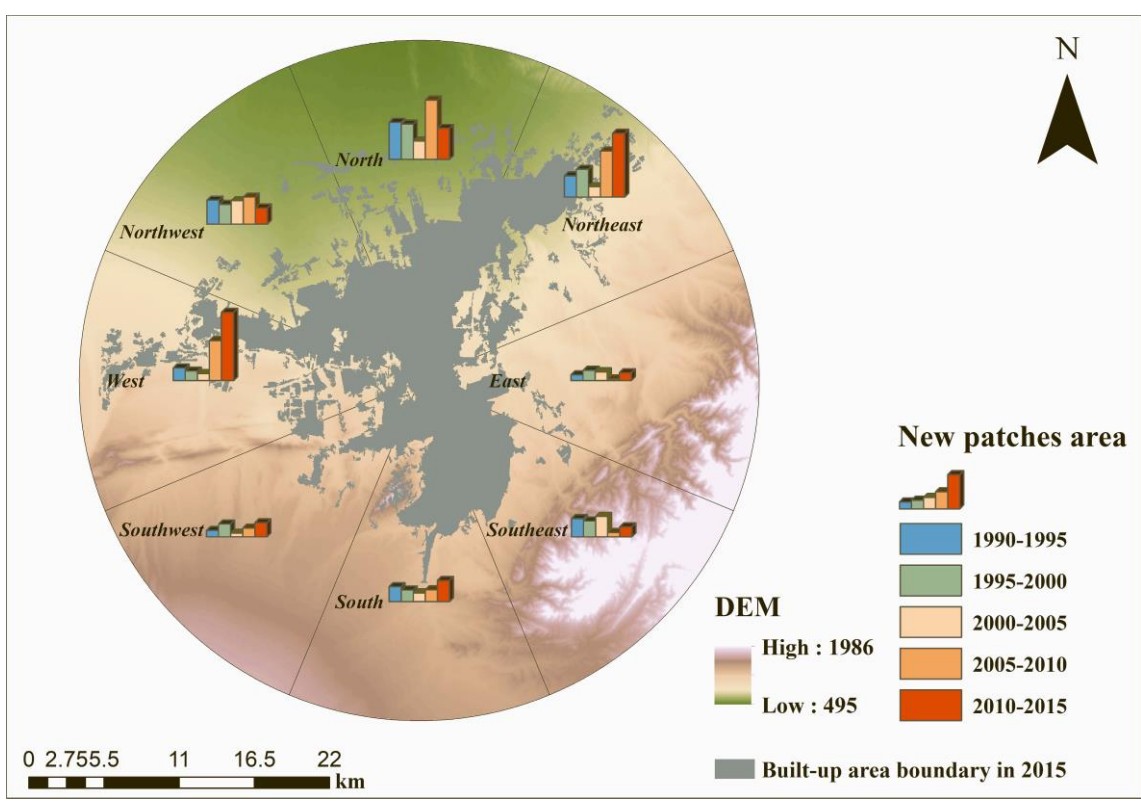

**Figure 7.** Eight-directional map of the urban expansion area of Urumqi from 1990 to 2015.

## 4. Discussion

Different from the reference-based *LEI*, which identifies landscape expansion patterns by empirical thresholds, this paper improves the extraction method of landscape expansion patterns with an enhancement of the exiting topological relationship such as ''overlap'', "touches", and "disjoint" among new patches, old patches and holes inside old patches, which corresponds to *infilling*, *edge-expansion* and *outlying*. Compared with the traditional

*LEI* (e.g., buffer method and envelope box method), the improved method is not affected by the shape of the patch and the parameters of the reference object. To a certain extent, it overcomes the problem of misclassification of landscape expansion patterns. The improved method was applied to Urumqi city, and the spatial and temporal distributions of landscape expansion patterns were extracted for five 5-year intervals between 1990 and 2015. The application results demonstrate that the area and number percentages of *edge-expansion* patches are always more significant than that of *outlying* and *infilling*, the number percentage of *outlying* is always larger than that of *infilling*, and the area percentage is lower than *infilling* only in 2000–2005. This indicates Urumqi's landscape expansion pattern of built-up land is dominated by *edge-expansion*, supplemented by *outlying*. Although not the dominant expansion pattern, the *outlying* patches provide new growth points for urban expansion. The proportion of *outlying* area was lower than *infilling* area during 2000–2005, probably due to the significant increase of the *outlying* patch area in the previous five years, which provided a large amount of excavatable land within the urban area, prompting *infilling* expansion to occupy these spaces further. According to the definition of landscape expansion pattern, *infilling* and *edge-expansion* belong to compact growth, and *outlying* belongs to diffuse growth. With the year 2000 as the boundary, the number proportion of *edge-expansion* patches showed a trend of decreasing and then increasing, precisely the opposite of *outlying*. In addition the number proportion of *infilling* patches showed a trend of increasing and then decreasing, with 2005 as the boundary. The proportion of the number of patches with diffuse growth rose between 1990 and 2000, inducing an increasing landscape fragmentation of urban built-up land. The proportion of the number of compact growth patches rose between 2000 and 2015, indicating a gradual decrease in urban landscape fragmentation and an increase in aggregation. The changing trend of *infilling* patches further indicates that the rate of development in urban interior space weakened during 2005–2015. The above trend is more evidence in the area of different expansion pattern patches. Based on the above analysis, we further summarize the fragmentation of landscape expansion patterns. As a dynamic quantitative model of urban growth patterns, the identification of the landscape expansion index can also be used to explain landscape fragmentation. Previous studies proposed a widely used model to quantify the fragmentation of forest landscapes, defining six patterns of landscape expansion with increasing fragmentation, including interior, perforated, edge, transitional, and patch [37,38]. The three urban landscape expansion patterns are a streamlining of the above six patterns and also represent a gradual increase in landscape fragmentation. *Infilling* is surrounded by built-up land patches, so it is not fragmented. The *edge-expansion* is adjacent to construction land and non-construction land patches representing a small degree of fragmentation. The *outlying* is surrounded by non-construction land patches and has the highest disturbance to the landscape and the greatest fragmentation. In summary, analyzing the results of landscape expansion patterns provides a quick understanding of urban development patterns and landscape fragmentation trends and also provides guidelines for the macro adjustment and layout of urban development in the next stage, ultimately reasonably avoiding the disorderly sprawl of the city.

The *MCI* and *AWCI* can reflect the compactness of the city at the landscape scale, which compensates for the problem that the topology-based method cannot quantitatively serve to estimate the compactness of urban expansion. Meanwhile, it can validate whether landscape expansion pattern identification results are consistent with the rule of urban morphological change. The results of *MCI* show that the urban expansion of Urumqi took 2000 as a turning point, presenting a scattered growth to a gradually compact expansion pattern. The changing trend of its response is consistent with the identification results of the three landscape expansion patterns, which validates the reliability of the landscape expansion pattern results. The *AWCI* is significantly influenced by the primary patches; thus, the sharp rise in *AWCI* from 1995 to 2000 and the decline in *MCI* indicate that the rapid development of the city during the early years caused the primary patches to become compact in advance of the whole city, while the landscape fragmentation of the entire city

also increased. The compactness of the maximum connectivity patches, the change in the area, and the area percentage also supports the tendency of the primary urban patches to be compact in advance. Correspondingly, *infilling* and *outlying* patches increased during this period, which is related to the existence of sufficient available land inside the urban area and the fragmentation of external expansion patches. In summary, *MCI* and *AWCI* can reflect the change of urban compactness more obviously and quantitatively at the landscape scale and complement the reliability of the extracted results of landscape expansion patterns. The quantitative metric provides decision makers with an intuitive overview of urban development compactness and allows them to learn from the development patterns of developed cities based on the comparison of landscape scale compactness changes in different cities.

Urban growth phase theory argues that urban development consists of two phases [39]: diffusion and coalescence. Corresponding to three landscape expansion patterns; *outlying* can be considered a diffusion process [18], *edge-expansion* and *infilling* can be regarded as a process of coalescence. The urban development level was lower in 1990, with the *LPI* being the weakest of any period, indicating that the city has no significant growth core. The relatively high *MCI* and *AWCI* further suggest that the city has not yet begun to expand significantly. The period of 1990–1995 was the initial stage of urban development; the rapid increase of *outlying* patches and *infilling* patches provided a large amount of new land for the city, while both *MCI* and *AWCI* decreased significantly, indicating that this stage corresponded to the diffusion stage of rapid urban development. From 1995 to 2000, the increment of *outlying* patches increased significantly. In addition, *MCI* dropped to the bottom and *AWCI* increased rapidly, indicating that the primary urban patches tended to aggregate, but the overall landscape fragmentation of the city was still increasing. Thus, this stage is still in the diffusion stage. During 2000–2005, the rate of urban expansion slowed down, the area of *outlying* patches decreased significantly, and the increment of *infilling* and *edge-expansion* remained basically unchanged. In addition, both *MCI* and *AWCI* increased. Therefore, the external expansion of the city slowed down and began to excavate available space internally, indicating a gradual transformation into a coalescence phase with reduced landscape fragmentation. The gradual increase in *MCI* between 2005 and 2015 indicates that the entire city tends to aggregate. The *AWCI* decreases slightly but is still at a high level, combined with the gradual decrease in the increment of *infilling* patches; this indicates that the primary patches are still in a compact development morphology. However, the space inside the built-up area is developed, and the city needs a new growth core. The results of landscape expansion pattern identification and urban compactness calculation showed that the urban development of Urumqi fits the development process of "diffusion-coalescence". Such a phenomenon is also experienced in the Nanjing [40], Dongguan [18] and Huangpi district of Hubei Province [41]. From the perspective of landscape pattern, the landscape pattern of the construction land exhibits a reduction in landscape fragmentation as the city grows.

The results of construction land transformation trajectories in the main urban area of Urumqi over the past 25 years reveal that the study area has experienced a dramatic evolution of the landscape pattern. The increase and persistence of the built-up land cover dominates the reconstructed land transfer matrix. The reason for this is that urban development is generally recognized as an irreversible process, that is, there is no de-urbanization of existing urban areas [39]. However, it can be observed that there is still a loss of construction land. This may be due to the urban redevelopment process, including the transformation of old buildings to demolished land, with the aim of increasing the efficiency of land use within the city [28]. Previous studies have shown that China has undergone massive urban redevelopment in the past few decades [42,43]. To gain an in-depth understanding of the directions and driver factors of urban expansion, we further analyzed the area of built-up land expansion in eight major directions. The expansion direction of Urumqi in the 25 years shows a general development pattern of "south control, north expansion, west extension first, then east expansion". The primary function of *infilling*

growth is to explore the potential of the inner city, and the expansion direction is mainly determined by the *edge-expansion* and *outlying*. The area of urban sprawl in the south, southwest and southeast districts is at a stable low value in all periods. At the same time, the basic absence of *edge-expansion* and *outlying* patch distribution can be observed, which is due to the higher elevation limiting urban sprawl. The Xinshi district and the Midong district in the north have experienced high rates of expansion and area in all periods and are the major distribution regions for *edge-expansion* and *outlying* patches. This is due to the extensive alluvial plain in the north that facilitates urban expansion and agricultural development. During 2000–2005, the northwest region took the lead as another growth pole and gradually extended to the west; after 2005 *edge-expansion* and *outlying* patches began to expand in large numbers to the Toutunhe district in the west. The eastern Shuimogou district is higher and farther away from the central urban zone. Thus, the expansion area is lower in all five periods. The northeast region has gradually sprawled around Midong into the third growth polarity of the city with the incorporation of Midong into Urumqi City after 2007. In summary, the direction of urban expansion in Urumqi is influenced by various factors, such as terrain, administrative district adjustment, macro layout of resources, etc. By analyzing the construction land transformation trajectory and expansion direction, we can understand the development pattern of the city and then reasonably constrain the *edge-expansion* and *outlying* patches to guide the efficient layout of urban space.

## 5. Conclusions

The classical landscape expansion index has the potential to capture the dynamics of the urban landscape, with limitations in identifying urban expansion patches showing as special shapes. In this paper, we modified the automatic identification method of landscape expansion patterns with topology theory, introduced the moment of inertia shape index to upgrade the method of shape-compactness, and validated these approaches using designed experiments and the application of the urban expansion evolution over Urumqi. The major conclusions are drawn and summarized as follows.

(1) The modified landscape expansion index we defined exhibits improved potential to capture the change process of urban landscape patterns accurately. The designed experiments demonstrated that both reference-based *LEIs*, the envelope box and the buffer methods, show apparent inaccuracy in identifying the type of urban expansion patches with special shapes, which our modified approaches can distinguish. The introduction of the topological relationship into the modified approaches provides the chance to effectively avoid misidentification of patches with special shapes.

(2) We defined *MCI* and *AWCI* in terms of the moment of inertia shape index to improve the capture of urban compactness within dynamics of urban expansion. The qualitative analysis and application over Urumqi indicate that the compactness trend is consistent with the trend of landscape expansion patterns; *MCI* and *AWCI* can catch the variation of compactness at the landscape scale for urban dynamics with a stable and high-quality evaluation.

(3) Both modified *LEI* and compactness indexes are applied in the urban expansion of Urumqi using data from the period 1990–2015. Our results show it is dominated by *edge-expansion* compact growth with secondary *outlying* growth. The results of *MCI* imply an elevated divergence and then convergence of urban compactness over Urumqi, with the turning point of year 2000, and *AWCI* indicates an earlier coalescence of main urban patches, with the direction of expansion influenced by factors such as terrain, administrative zoning policies and resource layout. This is consistent with the "diffusion-coalescence" development pattern of urban growth phase theory. After several decades of dramatic urban expansion, Urumqi has formed a basic pattern of "south control, north expansion, west extension first, then east expansion", as a "T" shape with a long north–south direction and a narrow east–west direction. This case application implies the potential of the modified approaches in identifying patterns of urban expansion patches and capturing the compactness of urban dynamics.

**Author Contributions:** Conceptualization, Y.T. and Y.S.; methodology, Y.T. and Y.S.; data organization, Y.T., L.T. and T.L.; writing—original draft preparation, Y.T. and Y.S.; writing—review and editing, X.M. and C.S.; supervision, Y.S.; funding acquisition, Y.S. All authors have read and agreed to the published version of the manuscript.

**Funding:** This research was supported by the National Key Research and Development Program of China (no. 2020YFA0608501), Talent recruited program of the Chinese Academy of Science (no. Y938091), National Natural Science Foundation of China (no. 42071351), and the project-supporting discipline innovation team of Liaoning Technical University (no. LNTU20TD-23).

**Data Availability Statement:** The standardized urban built-up area dataset (SUBAD) from 1990–2015 is available at (http://www.doi.org/10.11922/sciencedb.j00076.00004), accessed on 1 August 2022. The rest of the data presented in this study are available on request from the corresponding author.

**Acknowledgments:** We thank the relevant teams and organizations for providing the datasets used in this study. The standardized urban built-up area dataset (SUBAD) was provided by Science Data Bank and is available at http://www.doi.org/10.11922/sciencedb.j00076.00004/, accessed on 30 July 2021. The General Land Use Plan of the City of Urumqi (2006–2020) was provided by the Bureau of Statistics of the Government of Urumqi and is available at http://www.wlmq.gov.cn/. The Python code for calculating shape compactness of geographic features was provided by Lee Hachadoorian and is available at https://github.com/leehach/geocompactness/, accessed on 30 January 2022.

**Conflicts of Interest:** The authors declare no conflict of interest.

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
