# Peer review of "Improved Landscape Expansion Index and Its Application to Urban Growth in Urumqi"

_remotesensing, doi:10.3390/rs14205255_

Round 1

Reviewer 1 Report

This article is devoted to the practical topic of assessing dynamic changes in urban sprawl. The authors have done extensive work on data analysis, mapping and statistical processing, applying an improved method of landscape expansion pattern identification to capture expansion patterns and morphological changes in urban patches. Interesting results were obtained, which are useful to optimize territorial management.

However, the manuscript contains some very serious inconsistencies that need to be corrected prior to publication.

The first and perhaps the main shortcoming is the obscure English language: excessively long sentences that confuse the mind, inaccurate punctuation, obvious errors in the text, and a large number of superfluous and inappropriate words that make it difficult to understand phrases. In addition, there are some inconsistencies between the figures and the descriptions in the text (perhaps also due to translation flaws).

Secondly, the abstract section written by the author lacks examples of previous studies related to the city of Urumqi and is almost a list of other scholars' findings, which is not very meaningful. The abstract section should be highly condensed for the manuscript, presenting the highlights of the article, highlighting the key points, and briefly stating the significance of the study.

The "Introduction" section is too much text, and it can introduce the research related to urban dynamic change, and landscape extension pattern identification and then capture urban patches, and summarize its advantages and disadvantages, and finally state the reasons why the authors chose Urumqi as the study area, and the results obtained from the manuscript are of practical significance. What are the practical implications of the results obtained from this manuscript. The "Introduction" section that the authors are currently writing is too long and it is recommended that it be streamlined. The last paragraph of this section: lines 156-172 is too cumbersome and does not reflect the specific differences between the three different landscape extension indices compared, but only shows them.

In the "Materials and Methods" section, the author should write in the order of the process framework listed, firstly, data and pre-processing: the type of data and pre-processing part used in the manuscript should be briefly described; secondly, the methodological demonstration of the landscape expansion model; and finally, the methodological description of the landscape level compactness. methodological description of the degree of landscape level compactness. In the order in which the authors have listed them so far, it is confusing to know exactly what methods the authors want to express for the expansion and change of urban plates at the landscape level.

In this section, there is no diagram of the study area selected by the authors, and there is no brief description of the level of natural landscape and socio-economic development of the study area.

In the "Results" section, the descriptions in lines 296-300 are more like an overview of the study area selected in the manuscript and do not fit the results section; the descriptions in lines 291-306 should be more suitable for the data descriptions in the previous section "Data and Preprocessing". "Data and preprocessing" section of the data description; secondly, why did the authors only monitor the urban development of Urumqi city from 1990-2010, and why did they not extend it to 2020?

The data described in line303 is The standardized urban built-up area dataset of Urumqi city from 1990-2010, while the general land use plan of Urumqi city referenced in line305-306 is from 2006-2020, is there any inconsistency in the data years?

Please ask the authors to reconsider whether line290-306 should adjust the chapter order, this section is not stating the research results, but only an explanation of the data used in the manuscript and the policy reference basis, please write carefully.

The capture of Figure 4 is not clear and it is difficult to distinguish the extent of urban expansion in the figure for each year, and the authors are advised to enlarge Figure 4.

Table 7 is not annotated with units.

The discussion section suggests adding sections on why the results are what they are, why rapid changes are observed in one location and slow changes in another, what are the new results/methods compared to other similar studies, how are the results different and why, what do these results mean for management, what are the different approaches to their management interpretation, etc.

The discussion section essentially repeats these results. It is hoped to see here, in addition to a shorter presentation of the results, a more detailed proposal for their application in territorial planning.

Author Response

Thank you for your suggestions, we have merged all of them into one file, please see the attachment.

Reviewer 2 Report

The study has improved the landscape expansion index and studied the spatial and temporal evolution characteristics of urban expansion in Urumqi based on the improved landscape expansion index, which is innovative and has some research value. However, there are still several shortcomings in the study, and the specific recommendations are detailed in 1-14.

1. There are some typing and punctuation errors in the text, e.g. ArcGIS, line 196.

2. In lines 65-69, it is suggested to add the main content of this study, i.e. to specify the purpose and content of the study. The existing description only emphasises the importance of the urban expansion process, while the main subject of the study is the identification of urban morphological expansion, and the existing description still needs further improvement.

3. Line149, it is suggested to add a review of the progress of the existing research on the spatial indices of dynamic changes in urban expansion.

4. Lines 169-172, suggesting to add the significance of this article's research.

5. Lines 201-207, it is suggested to add a new title as a separate section.

6It is suggested to add an overview of the study area, data sources and data processing, and to add the reasons for selecting Urumqi as a case study.

7. Lines 290-301, the study content is suggested to be placed in the introduction.

8. What is the reason for choosing 1990-2010 as the study period? It is now 2022 and the timeliness of the research data needs to be improved.

In addition, the study results repeatedly describe that urban sprawl peaked in 2010, does it mean that cities are not expanding after 2010, or does it refer to the peak of 1990-2010? It is suggested that the description be improved to reflect the rigour of the language.

9.Lines 352-387, the elaboration of the content of Table 4, suggest adjusting the position to maintain the correspondence between the figure and the text.

10.Lines 388-389, what is the significance of Table 5? The description in the text is not fully reflected, suggesting additional explanation.

11.Recall the text, formulas 2-1, 2-2 corresponding to the table, graph, text description is? Tagging is recommended

12What are the units in Table 7? Also, it is suggested that the decimal places be kept consistent and that totals be added.

13.Lines 432-437, where improvements to the landscape extension index are suggested to be added in detail.

14.Lines 520-556 , the study conclusions are suggested to be appropriately streamlined.

Author Response

(The authors gave the same response as above.)

Reviewer 3 Report

Paper provides a solid application for detecting the type of urban increment through an improved landscape expansion index. Paper can be categorized as a GIS application paper instead of RS paper.the paper has potential but needs important revisions:

 1. Introduction is too long and the flaw is not good. Authors should separate Introduction to two parts and one should be a "Related Works" section. 

2. In line 95 Authors correctly stated that advances in RS helped dynamic urban monitoring but after two short sentences passed to the expansion patterns. RS related section should be enlarged with an overview of the current approaches for RS based urban monitoring with several citations.

3. Section 2 does not provide proper and detailed information about the materials used in this study (datasets) and the characteristics of the study area. Some information can be observed in further sections which also indicates the flaw problem of the paper.

4. Figure qualities are generally low.

5. Conclusion is too long and some of the content can go to Discussion.

Author Response

(The authors gave the same response as above.)

Round 2

Reviewer 1 Report

The manuscript was improved a lot, and it is better now. But there are still some problems, especially in data section. Update the dataset to 2020 and update the application results at the same time.

Reviewer 3 Report

Authors performed an extensive revision, which now reflects answers or modifications to my concerns. However, there is still an English quality problem throughout the paper and my advice is a complete read and review by a native speaker.
